# Altered resting-state functional connectivity of the frontal-striatal circuit in elderly with apathy

Chizuko Hamada[1]*, Toshikazu Kawagoe[2], Masahiro Takamura[3], Atsushi Nagai[3], Shuhei Yamaguchi[4], Keiichi Onoda[5]

1 Graduate School of Medicine, Shimane University, Izumo, Japan, 2 Liberal Arts Education Centre, Kyushu Campuses, Tokai University, Kumamoto, Japan, 3 Department of Neurology, Faculty of Medicine, Shimane University, Izumo, Japan, 4 Department of Neurology, Shimane Prefectural Central Hospital, Izumo, Japan, 5 Department of Psychology, Otemon Gakuin University, Oosaka, Japan

* okuzihc@med.shimane-u.ac.jp

**Data Availability Statement:** Data cannot be shared publicly because of the ethical policy because we did not explicitly denote that the data will be openly available in publication during

## Abstract

Apathy is defined as reduction of goal-directed behaviors and a common nuisance syndrome of neurodegenerative and psychiatric disease. The underlying mechanism of apathy implicates changes of the front-striatal circuit, but its precise alteration is unclear for apathy in healthy aged people. The aim of our study is to investigate how the frontal-striatal circuit is changed in elderly with apathy using resting-state functional MRI. Eighteen subjects with apathy (7 female, 63.7 ± 3.0 years) and eighteen subjects without apathy (10 female, 64.8 ± 3.0 years) who underwent neuropsychological assessment and MRI measurement were recruited. We compared functional connectivity with/within the striatum between the apathy and non-apathy groups. The seed-to-voxel group analysis for functional connectivity between the striatum and other brain regions showed that the connectivity was decreased between the ventral rostral putamen and the right dorsal anterior cingulate cortex/supplementary motor area in the apathy group compared to the non-apathy group while the connectivity was increased between the dorsal caudate and the left sensorimotor area. Moreover, the ROI-to-ROI analysis within the striatum indicated reduction of functional connectivity between the ventral regions and dorsal regions of the striatum in the apathy group. Our findings suggest that the changes in functional connectivity balance among different frontal-striatum circuits contribute to apathy in elderly.

## Introduction

Apathy is defined as a state of diminished motivation and goal-directed behavior, not attributable to decreased level of consciousness, cognitive impairment or emotional distress [1] Apathy occurs frequently in several neurodegenerative and neuropsychiatric disorders, and affects global cognitive function and clinical outcome [2].

Investigations of apathy mainly in neuropsychiatric diseases have been tried to explain its neural substrate with anatomical and functional alterations of brain circuits using different

informed consent. However, data are available from the corresponding author upon reasonable request. The institutional point of contact for this study is the Shimane University Institutional Committee on Ethics. Contact information (Email address) is kenkyu@med.shimane-u.ac.jp.

**Funding:** The authors received no specific funding for this work.

**Competing interests:** The authors have declared that no competing interests exist.

neuroimaging modalities. In Alzheimer's disease, apathy is associated with the atrophy of several brain regions, including the anterior cingulate cortex (ACC) and orbitofrontal cortex (OFC) [3]. Parkinson's disease with apathy showed dopaminergic denervation in the striatum, OFC, and posterior cingulate cortex [4], in addition to reduced gray matter density of the cingulate cortex and inferior frontal gyrus [5]. Other brain disorders with apathy such as stroke [6, 7], frontotemporal dementia [8], and depression [9] also showed anatomical and functional changes of the frontal region or striatum. Thus, the hypothesis that abnormalities within frontal-striatal circuits are relevant to apathy becomes prevailing.

The work of Levy and Dubois [10] proposed that symptoms of apathy can be divided into three subtypes of defects of processing; 'emotional', 'cognitive', and 'auto-activation'. These subtypes were assumed to correspond to the role of different regions of the frontal-striatal circuit; (1) 'emotional' is associated with the limbic territory including OFC, ventromedial prefrontal cortex (vmPFC), and ventral striatum, and (2) 'cognitive' is with the associative territory including dorsolateral prefrontal cortex (DLPFC) and dorsal caudate nucleus, and (3) 'auto-activation' is with the medial prefrontal cortex including ACC and supplementary motor area (SMA). Recently Le Heron et al. [11] adapted this framework to three stages of the normal motivative behavior based on abundant anatomical, biological and functional studies. First, the value system including the ventral striatum, vmPFC, and OFC is actuated. Then, the mediating system including the ventral striatum and ACC activates the motor system including the dorsal striatum, SMA, and posterior mid cingulate cortex, which achieves goal-directed behaviors.

Apathy is observed in older adults with normal cognitive function, and its prevalence is increased with aging and reduces their quality of life. Brodaty et al. [12] demonstrated apathy prevalence increased in a cognitively normal elderly cohort over a 5-year period from 6% to 15.8%. Moreover, presence of apathetic state in elderly is considered as an early sign of cognitive decline [13], and linked to lower activity [14]. Few neuroimaging studies have attempted to cope with apathetic state in healthy elderly adults, and it remains unclear whether the neural bases of apathy in normal aged people is the same as those in the neuropsychiatric groups. One study for aged population, structural changes of gray matter in the right putamen and the frontal and temporal lobes were related to apathetic state [15]. Moreover, the severity of apathetic state during cognitive tasks was related to reduced activation in the medial superior frontal gyrus, and changed activity in the DLPFC and left striatum [16]. Bonnelle et al. [17] also described that the ACC might play an important role in apathetic behavior of healthy people using task-related fMRI analyses. These findings imply apathetic state in healthy subjects is also likely associated with dysfunction of the frontal-striatum circuit.

Resting-state functional MRI (rs-fMRI) delineates functional connectivity between distant brain regions based on synchronized low-frequency fluctuations of blood oxygen level dependent (BOLD) signal [18]. Rs-fMRI has been widely used in both healthy subjects and patients with various neurogenetic and neuropsychiatric disorders. In several rs-fMRI studies of apathy in patients with Parkinson's disease [19] and Alzheimer disease [20] demonstrated reduction of functional connectivity among regions of functional connectivity of the frontal-striatal circuit. One rs-fMRI study of cognitively- normal aged people indicated decreased functional connectivity between the ventral striatum and frontal region in apathetic state [21]. These studies suggested that rs-fMRI is useful in assessing apathetic state in elderly.

Recent investigation of rs-fMRI about the frontal-striatal circuit have demonstrated that the segregated frontal-striatal loops influenced each other and information converges across the loops [22]. Haber [23] described anatomical evidence that communication across functionally distinct subregions of the striatum may occur for integrating information from parallel frontal-striatal functional modules for the development of goal-directed behaviors. It is presumed

that change of the communication across subregions of the striatum is observed in the diseases affecting the frontal-striatal circuit. For example, Bell et al [24]. demonstrated the reduction of functional connectivity across striatal subdivision in Parkinson's disease compared to healthy control.

In this study, we applied region of interest (ROI)-based analyses for detecting distinct patterns of functional connectivity with/within the striatum. Twelve seeds placed throughout the striatum, six in each hemisphere, described by Di Martino et al. [25] were used. These seeds were derived from previous studies of anatomical and functional subdivisions of the striatum, and each seed was related to classical parallel and integrative frontal-striatal loop [26]. We aimed to investigate the underlying mechanism of apathy related to disfunction of the frontal-striatum circuit in healthy subjects by the analyses with the multiple seeds in the striatum, and located the alterations of interaction and organization among the different frontal-striatum circuits in the elderly with apathy.

## Materials and methods

### Participants

We recruited subjects who voluntarily participated in the brain checkup system at Shimane Institute of Health Science, Izumo City, Shimane Prefecture, Japan. The health check included physical examination, detailed medical history, laboratory blood tests, neuropsychological assessment, and cranial MRI. A total of 335 people in the database were screened on their apathy levels and the exclusion criteria. Exclusion criteria were: current or past presence of neurological or psychiatric disorders; Mini-mental State Examination (MMSE) score less than 26 or dementia; obvious structural brain disorders; and head motion $> 2.0$ mm during the scan of rs-fMRI. Clinical MRIs were evaluated by a trained neurologist and neuroradiologist.

Finally, 18 people with apathy (7 females, age: $63.7 \pm 3.0$ years, rage 61–68) and 18 people without apathy (10 females, age: $64.8 \pm 3.0$, years, rage 60–69) matched for age, gender, and years of education were included. The study was conducted in accordance with the Declaration of Helsinki (1975, as revised in 2008) and the regulations of the Japanese Ministry of Health, Labour and Welfare, and approved by the ethical committee of the Shimane University School of Medicine. All subjects gave written informed consent to this study.

### Neuropsychological and neuropsychiatric measures

Apathy level was evaluated with the Japanese version of the Apathy Scale (AS) [27, 28]. This scale consisted of 14 items and was used in a self-assessment style. The AS ranges from 0 to 42 and higher AS values indicate higher apathetic status. The cutoff point was determined on the basis of the previous report on Japanese stroke patients, and the scale displayed a high validity [28] (sensitivity 81.3%, specificity 85.3%) with a cutoff point of 16 among the participants. In this study, the apathy group was composed of 18 participants with the AS score of 16 point or more, and the participants in the non-apathy group had the AS score less than 16.

Furthermore, all participants conducted the following neuropsychological assessments: MMSE [29], Frontal Assessment Battery (FAB) [30], and Kohs Block Design Test (KOHS) [31]. Depressive symptoms were evaluated using the Japanese version of Zung's self-rating depression scale (SDS) [32].

### Image acquisition

Imaging data were acquired using a Siemens AG 1.5 T scanner. Using $T2^{*}$-weighted gradient-echo spiral pulse sequence, we measured twenty axial slices parallel to the plane connecting the

anterior and posterior commissures. (repetition time = 2000 msec, echo time = 35 msec, flip angle = 90˚, scan order = interleave, matrix size = 64 × 64, field of view = 256 × 256 mm², isotropic spatial resolution = 4 mm, slice thickness = 5 mm, gap = 1 mm). We underwent 5-minute rs-fMRI scan after it was directed that all subjects stayed awake and closed their eyes. After the functional scans, we measured T1-weighted images of the whole brain (192 sagittal slices, repetition time = 2170 msec, echo time = 3.93 msec, inversion time = 1100 msec, flip angle = 90˚, matrix size = 256 × 256, field of view = 256 × 256 mm², isotropic spatial resolution = 1 mm).

## Data preprocessing

SPM12 (https://www.fil.ion.ucl.ac.uk/spm/software/spm12/, RRID:SCR_007037) was used for MRI data preprocessing. The first 10 functional images of each subject were discarded for magnetic field stabilization. The remaining 140 functional images were realigned to remove any artifacts from head movement and to correct for differences in image acquisition time between slices. Average of max head motions were 0.29 ± 0.19 mm for the apathy group and 0.30 ± 0.20 mm for the non-apathy group. Next, the functional images were normalized to the standard space defined by a template T1-weighted image and resliced with a voxel size of 3 × 3 × 3 mm³ to adjust the gray matter probability maps. After spatial smoothing was applied with the FWHM equal to 8 mm, temporal preprocessing was performed with the CONN toolbox (https://web.conn-toolbox.org/, RRID:SCR_009550). The denoising steps included regressions of six bulk motion parameters and their first-order derivatives, the five potential noise components estimated from the white matter and cerebrospinal fluid [33], the scrubbing outlier scans were identified based on ART with intermediate settings (97 percentiles in normative sample), and band-pass filtering of 0.008–0.09 Hz.

## Selection of region of interest (ROI)

We adopted 12 striatum seeds, which were delineated by Di Martino et al [25]. Each seed was as follows: the inferior ventral striatum (VSi) (± 9, 9, -8), superior ventral striatum (VSs) (± 10, 15, 0), dorsal caudate (DC) (± 13, 15, 9), dorsal caudal putamen (DCP) (± 28, 1, 3), dorsal rostral putamen (DRP) (± 25, 8, 6), and ventral rostral putamen (VRP) (± 20, 12, -3). The radius of each ROIs was 3.5 mm. The bilateral pairs of seeds were combined into a single ROI in ROI-to-voxel analysis. In ROI-to-ROI analysis, all 12 seeds were independently considered as ROIs.

## Functional connectivity analysis

Data analyses were carried out using the CONN toolbox. In the first-level analysis, we obtained ROI-to-voxel functional connectivity maps for each individual subject. This analysis produced z-score maps of positive and negative correlation coefficients for 6 striatal seeds and whole-brain voxels, combining homologous left and right ROIs to one seed. We also conducted an ROI-to-ROI analysis to examine functional connectivity between 12 ROIs within the striatum for each subject.

In the second-level analysis, at first, we conducted one-sample t-test of whole brain functional connectivity for 6 striatal ROIs across 36 participants to acquire the functional connectivity map. Then, to assess the differences in functional connectivity between the apathy and non-apathy group, we conducted two-sample t-test controlling for confounding factors including age and sex with the statistical threshold at the voxel level of p < 0.001 and the cluster level of p < 0.05 for false discovery rate (FDR). Moreover, we added a group comparison of the ROI-to-ROI analysis for the purpose of examining functional connectivity among 12

striatal ROIs with FDR correction of p < 0.05. Finally, we conducted similar group comparisons of ROI-to-voxel and ROI-to-ROI using SDS as an additional covariate, because apathy and depression share common clinical features and they show high correlation among neuropsychological tests.

## Results

### Demographic data

A total of 36 (17 female) healthy participants (mean age; 64.2 years, SD; 3.0) were assessed for study eligibility. We assigned 18 subjects to the apathy group and 18 subjects to the non-apathy group. Table 1 summarizes our subjects' baseline demographic and neuropsychological data. Statistical analyses of demographic and neuropsychological data were performed using two-sample t tests. Fisher's exact test was used to compare gender distribution across the samples. P values less than 0.05 were considered significant. There were no significant differences between the apathy and non-apathy group in age, sex, educational status, and cognitive test scores (including the MMSE, FAB, and KOHS). The apathy group's mean depression score was higher than the non-apathy group (t = 6.18, p < 0.01).

### Functional connectivity

The functional connectivity map of 6 striatal ROIs across 36 participants was similar to a previous study [25] that demonstrated connectivity between striatum subdivisions and other cerebral regions such as the frontal cortex, parietal cortex, and temporal cortex. Fig 1A shows functional connectivity maps among the VRP, DC and whole brain, and comparisons between the apathy and non-apathy groups. The distribution of functional connectivity for each seed of striatum were similar, but slightly different, for each seed. We performed two-sample t-test to explore differences in whole brain functional connectivity between the apathy and non-apathy groups. The apathy group showed decreased functional connectivity between the bilateral VRP and the cluster of voxels in the right dorsal anterior cingulate cortex/ pre supplementary motor area, (dACC/pre SMA) (x = 9, y = 6, z = 39, voxels = 48, p = 0.008 for FDR) compared with the non-apathy group. Moreover, the functional connectivity between bilateral DC and the cluster of voxels in the left sensorimotor area (x = -51, y = -15, z = 42, voxels = 63, p = 0.0005 for FDR) was increased for the apathy group compared with the non-apathy group (Table 2). There were no significant group differences in the analyses of other seeds.

**Table 1. Demographic and clinical characteristics of the apathy and non-apathy groups.**

|                   | Apathy (n = 18) | Non-apathy (n = 18) | Statistics (p-value) |
|-------------------|-----------------|---------------------|----------------------|
| Age (years)       | 63.7 (3.0)      | 64.8 (3.0)          | 0.276                |
| Sex (female)      | 7               | 9                   | 0.738                |
| Education (years) | 12.2(2.0)       | 13.4(2.9)           | 0.148                |
| MMSE              | 28.3 (1.7)      | 28.9 (1.4)          | 0.302                |
| FAB               | 15.9 (1.8)      | 16.4 (1.3)          | 0.309                |
| KOHS              | 99.1 (19.9)     | 105 (13.9)          | 0.277                |
| AS                | 19.2 (2.1)      | 2.6 (1.7)           | <0.001               |
| SDS               | 38.6 (5.0)      | 28.2 (5.1)          | <0.001               |

Note. Data are demonstrated as means (standard deviation). Abbreviations: FAB, Frontal Assessment Battery; KOHS, Kohs Block Design Test; AS, Apathy scale; SDS, Zung's self-rating depression scale.

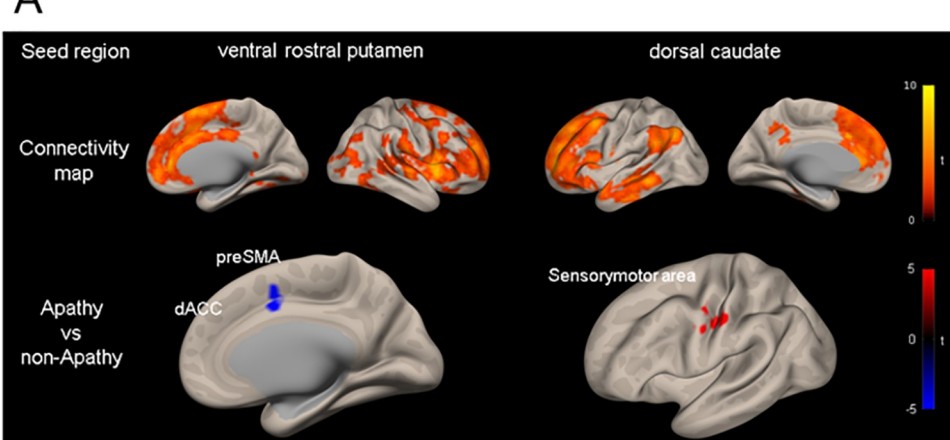

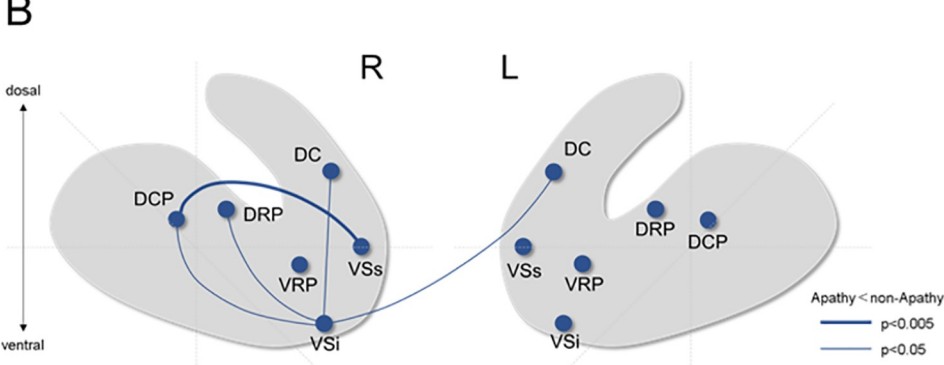

**Fig 1. Changes of resting-state functional connectivity associated with apathy.** (A) Seed-tovoxel analysis. The upper row shows whole brain maps of functional connectivity with seeds of the VRP and DC. The lower row shows difference of the functional connectivity between the apathy and non-apathy group for the seeds of VRP and DC. Red and blue foci indicate areas showing significant positive and negative connectivity differences, respectively. (B) Group comparison of ROI-to-ROI analysis for 12 striatal seeds. The blue lines indicate decreased connectivity in the apathy group.

To investigate the effect of apathy on the laterality of functional connectivity between the basal ganglia and cerebral cortex, we conducted a two-way ANOVA with GROUP (apathy/non-apathy) and HEMISHPERE (the bilateral pairs of seeds), and the result did not show significant difference between the right and left striatum seeds.

We performed ROI-to-ROI analysis within the striatum and compared the functional connectivity between two groups (Fig 1B). Compared with the non-apathy group, the apathy group showed greater reduction of functional connectivity between the right VSs and right DCP (t = -3.22, p = 0.016 for FDR). Functional connectivity between the VSi and dorsal

**Table 2. Regions showing significant functional connectivity differences between groups.**

| seed | | region | voxels | cluster size p-FDR | MNI coordinates | | |
|---|---|---|---|---|---|---|---|
| | | | | | x | y | z |
| ventral rostral putamen | A>NA | R dACC/pre SMA | 48 | 0.008 | 9 | 6 | 39 |
| dorsal caudate | A>NA | L sensorimotor | 63 | 0.0005 | -51 | -15 | +42 |

Abbreviations: A, Apathy; NA, non-Apathy; dACC, dorsal anterior cingulate cortex; pre SMA, pre supplementary motor area.

striatum was also reduced (ts < -2.35, ps < 0.042 for FDR) in the apathy group. Moreover, functional connectivity between the right VSi and left DC (t = -2.45, p = 0.0420 for FDR) was reduced after controlling for SDS to control for the effects of depression (t = -2.68, p = 0.06 for FDR).

## Discussion

In the current study, we found that people with apathy showed decreased functional connectivity between the VRP and dACC/pre SMA, along with increased functional connectivity between the DC and left sensorimotor area compared with people without apathy. In addition, the functional connectivity between some of 12 striatal ROIs was decreased for the apathy group compared with the non-apathy group. These changes were observed after controlling for depression score. Our results indicate that functional disruption of frontal-striatal circuits is associated with apathetic state in the elderly, independent of depression.

The dACC has been considered to be an important component of the classic frontal-striatal circuit model such as the parallel loops model [34] and the recent interactive cortico-striatal system [35]. The dACC, which is also called anterior midcingulate cortex, [36] is specialized for monitoring of errors [37] and conflict [38], evaluating choice behavior, and regulation of an action [39]. Cumulative evidence suggests that the dACC represents a hub where information about reinforcers is linked to guide appropriate action. Consequently, appropriate activation of dACC results in self-generated movement or goal-directed behavior. Actually, the several previous studies demonstrated that the dysfunction of dACC is associated with apathetic state in healthy people [40]. Recent conceptual framework of cortico-striatal circuit have speculated that the connections between functionally different frontal areas involved in emotion, cognition and motor function and the striatum are segregated and overlapped, and their interactions enable to carry out goal-directed actions [41]. The projections of the striatum are topographically organized, such that the ventromedial area of striatum is connected to the limbic frontal area, the dorsolateral area of striatum is connected to the motor-related frontal area, and the intermediate area of striatum, to the dACC and DLPFC, which is related to cognitive process. Our study showed that people with apathy had decreased functional connectivity between the dACC and VRP which is corresponded to the ventromedial area of striatum. These results support the notion that apathetic state in aged people is associated with defects of emotional and/or cognitive mechanisms in the frontal-striatal circuit. One recent rs-fMRI study also revealed that apathetic state was associated with reduction of functional connectivity between the ventral striatum and dACC [42].

The SMA is considered as a part of the motor-related areas on the medial surface of cerebral hemisphere [43], which plays an important role in generation and control of movement. The pre SMA that is separated from classical SMA [44] is located in the rostral portion of Brodmann area 6. The pre SMA is involved in more complex motor and cognitive process associate with goal-directed behavior [45], whereas the SMA is more closely related to movement execution [46]. The anatomical and functional studies demonstrated that the pre SMA is connected to the premotor area, dACC, and DLPFC but not motor area [47], and is also connected to intermediate portion of the striatum [48, 49]. Thus the pre SMA was considered to be a component of cognitive system [50, 51]. To the best our knowledge, this is the first rs-fMRI study that demonstrated decreased functional connectivity between the VRP and pre SMA in people with apathy. This result implies that the VRP located in the relatively dorsal part of ventral striatum is more related to cognition than the VSi located in the ventral part of ventral striatum and generally used as a ROI of the ventral striatum.

Another finding of the present study was the reduction of functional connectivity among ROIs within the striatum in the apathy group compared to the non-apathy group. We found that functional connectivity was more severely impaired between the VSs and DCP. The DCP is situated in the most dorsolateral and caudal part among our defined ROIs and has connections to motor and premotor areas [41]. The VSs is located in the dorsolateral part of ventral striatum, and is considered as not only the region connected to limbic frontal area but also the region related to frontal area involved in cognition [52]. Previous anatomical and functional studies reported that the portion of ventral striatum corresponding to the VSs was projected from the lateral OFC, and was involved in reward-associated decision making [53], while the VSi was primarily associated with limbic areas including the medial OFC. Moreover our result showed that the VSi, which approximately corresponds to the nucleus accumbens, also showed reduction of functional connectivity with the dorsolateral regions of striatum connected with cognitive areas including DLPFC and dACC [25]. A few studies on apathy demonstrated anatomical and functional alterations within the striatum, for instance, diminished ventral striatum volume [54, 55] and decreased functional connectivity between nucleus accumbens and another striatum area [42], but these studies have not focused on direct connectivity between individual subdivisions of the striatum. Our result, together with other studies, suggests that apathetic state may be attributed to changes of the balance between dorsal and ventral striatum network.

Furthermore, the current study demonstrated increased functional connectivity between the DC and sensorimotor area in people with compared with people without apathy. The DC is located in dorsal area of the striatum, and is connected to cognitive prefrontal areas like dorsolateral prefrontal cortex. Several rs-fMRI studies for apathy demonstrated hyperconnectivity between the DLPFC and the superior parietal cortex in Alzheimer disease [20], between the superior frontal gyrus and the thalamus in Parkinson's disease [56]. These regions are included in the salience and attentional/executive systems and their hyperconnectivity was speculated as compensatory phenomenon for dysfunction of other frontal-striatal systems. Our result also suggests that functional dysfunction in the cognitive system in apathy could activate motor function area for compensation.

We analyzed functional connectivity using the SDS sore as a covariate to minimize the effect of depression. Apathy and depression are often overlapped clinically, and their diagnostic criteria are partly common [57]. But recent studies indicated that frontal-striatal circuit of apathy was separated from the network for depression. Several studies demonstrated that people with apathy had decreased functional connectivity associated with frontal-striatal circuit, whereas depressive people had increased functional connectivity with similar region reciprocally, such as between the dACC and other salience area [40], between caudate and thalamus, and between DFC and parahippocampal cortex [58]. In the present study, only when ROI-to-ROI analysis within the striatum was performed, significant difference in functional connectivity in the apathy group survived after controlling for SDS.

We have some limitations in the current study. We examined apathetic state using the AS, but the scale was not configured for classification of apathy subtype, and some questionnaire items overlapped with the SDS. A multidimensional scale has been recently developed to assess the subtypes of apathy [59]. Using such an assessment tool, we might delineate the various frontal-striatal circuits related to subtypes of apathy. Moreover, to ensure that the participants were clearly independent of depression, an apathy group without depression should have been included.

In addition, our subjects were aged people without neurodegenerative and psychiatric diseases, but apathy prevalence is lower in cognitively normal cohort than patient groups (such as, dementia) [60]. Specificity and similarity of apathetic state associated with healthy people

and diseases have been unclear. Further studies about apathetic state of healthy people should address this issue.

In conclusion, our findings showed that apathetic state in aged people was associated with altered resting-state functional connectivity with/within the striatum, specifically decreased functional connectivity of the striatum with emotional and cognitive frontal-striatal circuits, and between ventral and dorsal regions within the striatum. These results suggest that alteration of interaction and organization among different frontal-striatum circuits contributes to apathy in elderly.

## Acknowledgments

We would like to thank Editage (www.editage.com) for English language editing.

## Author Contributions

**Conceptualization:** Chizuko Hamada, Shuhei Yamaguchi, Keiichi Onoda.

**Data curation:** Chizuko Hamada.

**Formal analysis:** Chizuko Hamada, Toshikazu Kawagoe, Masahiro Takamura, Keiichi Onoda.

**Methodology:** Chizuko Hamada, Toshikazu Kawagoe, Shuhei Yamaguchi, Keiichi Onoda.

**Supervision:** Atsushi Nagai, Shuhei Yamaguchi, Keiichi Onoda.

**Writing – original draft:** Chizuko Hamada.

**Writing – review & editing:** Toshikazu Kawagoe, Masahiro Takamura, Atsushi Nagai, Shuhei Yamaguchi, Keiichi Onoda.

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
