## [Decision Letter · Decision Letter 0]

26 Jul 2021

PONE-D-21-12168

Altered resting-state functional connectivity of the frontal-striatal circuit in elderly with apathetic state

PLOS ONE

Dear Dr. Hamada,

Thank you for submitting your manuscript to PLOS ONE. After careful consideration, we feel that it has merit but does not fully meet PLOS ONE’s publication criteria as it currently stands. Therefore, we invite you to submit a revised version of the manuscript that addresses the points raised during the review process.

We look forward to receiving your revised manuscript.

Kind regards,

Satoshi Ikemoto

Academic Editor

PLOS ONE

2. Please consider rephrasing "apathetic people" to "patients/people with apathy", as our our submission guidelines (http://journals.plos.org/plosone/s/submission-guidelines) suggest changing potentially stigmatizing labels should be changed to more current, acceptable terminology.

3. We note that Figures A & B in your submission contain copyrighted images. All PLOS content is published under the Creative Commons Attribution License (CC BY 4.0), which means that the manuscript, images, and Supporting Information files will be freely available online, and any third party is permitted to access, download, copy, distribute, and use these materials in any way, even commercially, with proper attribution. For more information, see our copyright guidelines: http://journals.plos.org/plosone/s/licenses-and-copyright.

a. You may seek permission from the original copyright holder of Figures A &B to publish the content specifically under the CC BY 4.0 license.

4. We noticed you have some minor occurrence of overlapping text with the following previous publication(s), which needs to be addressed:

- https://direct.mit.edu/jocn/article/24/11/2186/27852/Decreased-Functional-Connectivity-by-Aging-Is

In your revision ensure you cite all your sources (including your own works), and quote or rephrase any duplicated text outside the methods section. Further consideration is dependent on these concerns being addressed.

Additional Editor Comments (if provided):

Reviewers' comments:

Reviewer's Responses to Questions

**Comments to the Author**

1. Is the manuscript technically sound, and do the data support the conclusions?

Reviewer #1: Partly

Reviewer #2: Partly

2. Has the statistical analysis been performed appropriately and rigorously? 

Reviewer #1: Yes

Reviewer #2: Yes

3. Have the authors made all data underlying the findings in their manuscript fully available?

Reviewer #1: Yes

Reviewer #2: Yes

4. Is the manuscript presented in an intelligible fashion and written in standard English?

Reviewer #1: Yes

Reviewer #2: No

5. Review Comments to the Author

Reviewer #1: The present study compared resting state striatal functional connectivity between the apathy and non-apathy groups. The connectivity between the ventral rostral putamen and the right dorsal anterior cingulate cortex/supplementary motor area was found decreased in the apathetic group while the connectivity between the dorsal caudate and the left sensorimotor area was increased. The ROI-to-ROI analysis revealed reduction of functional connectivity between the ventral regions and dorsal regions of the striatum in the apathy group. These results may help us to under stand the neural alteration associated with apathy. This reviewer has several concerns as below.

The description of data preprocessing on page 8 is not clear. The phrase of ‘scrubbing parameter with a band-pass filter of 0.01-0,08Hz’ is confusing. Considering the later sentence stated that the scrubbing parameter was used to identify the outlier scans, did the authors want to express in the first sentence that a band pass filtering was used and a scrubbing strategy was used to cutoff outlier scans using a ‘parameter’ generated by the toolbox? If so, what parameter and threshold was used to determine an outlier scan? For example, the frame-displacement (FD >0.5mm) was used in many previous studies.

The description of “functional connectivity analysis” is not clear. It was said that ‘In the first-level analysis, a multiple regression analysis was performed for each individual subject using the general linear model for correlation connectivity estimation’. My understanding of the procedure is that ‘the multiple regression analysis’ was used in the preprocessing stage to regress out nuisance variables including head motions and others. Then the resultant errors were band-pass filtered. The time course of each seed mask was extracted and correlated with the time course of each voxel. The correlation coefficients were then transferred with Fisher’s Z-transformation.

The FWHM of 8 mm seem too big as the seed ROIs are close to each other. In literature, 5 or 6mm is a typical FWHM and may be more proper for the current study.

The left and right homologous ROIs were combined to one seed when calculating functional connectivity maps. But the human brain laterization is well known. It’s better to conduct a 2-by-2 ANOA for the second level analysis with GROUP and HEMISHPERE as factors.

When reporting results, the functional connectivity of several seeds seems to have no difference between the two groups. But it was not explicitly stated.

The readability of Figure 2b is poor. I suggest a different font color or change the back groud color.

Reviewer #2: I understand from this manuscript that elderly individuals with apathy show decreased functional connectivity between the striatal structures (i.e., ventral rostral putamen), and frontal areas (i.e., dACC/pre SMA). Besides, increased functional connectivity between the striatal structures (i.e., dorsal caudate) and sensorimotor area was reported in elderly individuals with apathy as compared to elderly individuals without apathy.

Functional neuroanatomy studies are important for a better understanding of apathy which increases disability and caregiver burden in elderly population. This is a powerful study in terms of using fMRI analysis methods. I think if the authors strengthen the presentation of their study, it will contribute to the literature and colleagues of the field.

Major points:

1) The areas associated with apathy in AD are not just the ACC and OFC, or the entire ACC or OFC. If it is desired to exemplify some of the related brain regions, it may be more appropriate to reconstruct the sentence (Introduction section- Line 26).

2) 7th reference seems to be missing in the main text.

3) Extended sentences with conjunctions make it difficult to follow, and in some there may be errors in tense suffixes. In this sense, I would suggest that the manuscript be reviewed by the authors.

4) The authors may consider to edit the flow of Introduction section. The paragraph starting with "Apathy is also observed in older adults with..." from line 43 does not seem to be compatible with the previous paragraph. In the previous paragraph, the conceptualization of the apathy and the functional neuroanatomical correlates corresponding to this conceptualization are presented. However, the first sentence containing the word "also" causes the expectation for continuum of a topic.

5) I think, the first paragraph of the Materials and Methods/Participants section also need to be reconsidered. Especially these two sentences are hard to follow: "335 people in the database were screened on the levels of apathy and the exclusion criteria. Exclusion criteria were: current or past presence of neurological or psychiatric disorders such as cerebrovascular disease, Mini-mental State Examination (MMSE) score less than 26 or dementia, diffuse or multiple cerebral white matter lesions on T2-weighted image, obvious cerebral atrophy, head motion > 2.0 mm during the scan of rs-fMRI". Since this section contains basic information for readers to understand the study, I think it would be more useful to give it in simpler sentences.

6) It is not common to use cut-off points on apathy scales. In general higher scores indicate higher levels of apathy. Was the cut-off point for the apathy scale used in a single study? Are there any other examples in recent studies, supporting the validation of this cut-off score other than the relevant reference (#32)?

7) Explaining the scoring of depression scale will make it possible to understand the results and the table. Does this depression scale involves any cognitive tasks? If not, I would strongly recommend not to cluster depression scale under the neuropsychological assessment.

8) Line 151. Giving an analysis result and comparative reference information in the Method section may not be very appropriate. I recommend to relocate that sentence to Discussion.

9) Under the figure: A)... Red and blue foci indicate areas showing significant positive and negative connectivity.

It is better to clarify the color and connectivity direction.

10) Creating a table with peak Talairach coordinates for ROIs can make it easier for the readers to understand the results. It will also be convenient for the study to be repeated by other researchers.

11) The authors drew attention to the overlap in symptoms of apathy and depression. However, if apathy and depression comorbidly present; it may not be easy to draw a line between these two syndrome as we can do between an internal disease and a psychiatric condition. It is known that both depression and apathy are the neuropsychiatric conditions that have independent effects on the molecular structure and physiological functioning of the brain. Statistically controlling the depression score may not result with subtracting the effect of depression from the brain activation as intended here in this study. I find the study's functional imaging analysis approach correct (controlling depression by adding the scores to analysis as covariant). However, adding multi covariates in fMRI analysis may suppress the overall activation which can cause underestimating the findings. In my opinion, in order to be able to say that the results were clearly independent of depression, a group of apathetic participants without depression should have been included. I would recommend the authors to discuss their method and findings stronger, in this context. Also, if the researchers have used any other tools or measurements, those can be included to manuscript in terms of increasing internal consistency.

Minor points:

Line 41. Mistyping of striatum as stratum

Line 60. I would suggest using "study of cognitively normal aged people" instead of "study of cognitive- normal aged people"

Line 85. I would suggest using "cranial MRI" instead of "head MRI"

Line 104. This long sentence "Furthermore, all participants conducted the following neuropsychological assessments: Cognitive functions, assessed by the MMSE [29], the Frontal Assessment Battery (FAB) [30], and the Kohs Block Design Test (KOHS) [31]; depressive symptoms were evaluated using the Japanese version of Zung’s self-rating depression scale (SDS) [32]." can be simply expressed like "Furthermore, all participants conducted the following neuropsychological assessments: MMSE [29], Frontal Assessment Battery (FAB) [30], and Kohs Block Design Test (KOHS) [31]. Depressive symptoms were evaluated using the Japanese version of Zung’s self-rating depression scale (SDS)"

Line 126. Could the authors have meant "adjust" by using the phrase "to agree with" in this sentence: Next, the functional images were normalized to the standard space defined by a template T1-weighted image and resliced with a voxel size of 3x3x3 mm3 to agree with the gray matter probability maps.

Line 209. If the intended meaning is not the process of aging; authors may consider using "elderly" instead of using the expression "aged people".

6. PLOS authors have the option to publish the peer review history of their article (what does this mean?). If published, this will include your full peer review and any attached files.

Reviewer #1: No

Reviewer #2: No

---

## [Author Response · Author response to Decision Letter 0]

11 Oct 2021

Satoshi Ikemoto

Academic Editor

PLOS ONE

U.S. Headquarters

1265 Battery Street, Suite 200

San Francisco, CA 94111 United States

Re: Manuscript ID: PONE-D-21-12168

Dear Prof. Ikemoto,

Thank you very much for your e-mail and review of our manuscript (PONE-D-21-12168) entitled “Altered resting-state functional connectivity of the frontal-striatal circuit in elderly with apathy”. We also appreciate the reviewers taking the time to offer us your comments and insights related to the paper. In the following sections, you will find our responses to each of your points and suggestions.

Q1. Please ensure that your manuscript meets PLOS ONE's style requirements, including those for file naming. The PLOS ONE style templates can be found at

A1. We appreciate the helpful suggestion regarding Journal requirements. As suggested, we confirmed PLOS ONE's style requirements and corrected the whole manuscript accordingly.

Q2. Please consider rephrasing "apathetic people" to "patients/people with apathy", as our submission guidelines (http://journals.plos.org/plosone/s/submission-guidelines) suggest changing potentially stigmatizing labels should be changed to more current, acceptable terminology.

A2. We accept these journal requirements. The suggested key terms have been changed throughout the manuscript, title, and short title. 

Q3. We note that Figures A & B in your submission contain copyrighted images. All PLOS content is published under the Creative Commons Attribution License (CC BY 4.0), which means that the manuscript, images, and Supporting Information files will be freely available online, and any third party is permitted to access, download, copy, distribute, and use these materials in any way, even commercially, with proper attribution. For more information, see our copyright guidelines: http://journals.plos.org/plosone/s/licenses-and-copyright.

A3. Thank you for this comment. Figures A & B in our submission are original figures. While our figures (especially Figure B) are indeed inspired by a previous study (Bell et al., 2015), we present that the only similarity between the figures is in the shape of the basal ganglia, and that there is no copyright on this shape, which reflects a generally accepted fact. We also present that the way Figure A is presented is completely typical and common.

Reference

Peter Bell et al. (2015) Dopaminergic basis for impairments in functional connectivity across subdivisions of the striatum in Parkinson's disease. Hum Brain Mapp. 2015 Apr;36(4):1278-91. doi: 10.1002/hbm.22701

Q4. We noticed you have some minor occurrence of overlapping text with the following previous publication(s), which needs to be addressed:

-https://direct.mit.edu/jocn/article/24/11/2186/27852/Decreased-Functional-Connectivity-by-Aging-Is

In your revision ensure you cite all your sources (including your own works), and quote or rephrase any duplicated text outside the methods section. Further consideration is dependent on these concerns being addressed.

A4.

We thank the editor for pointing this out and also recognize that this point is something that needs to be improved in our paper. Accordingly, we have tried to paraphrase as much as possible.

Reviewers' comments:

Reviewer #1: The present study compared resting state striatal functional connectivity between the apathy and non-apathy groups. The connectivity between the ventral rostral putamen and the right dorsal anterior cingulate cortex/supplementary motor area was found decreased in the apathetic group while the connectivity between the dorsal caudate and the left sensorimotor area was increased. The ROI-to-ROI analysis revealed reduction of functional connectivity between the ventral regions and dorsal regions of the striatum in the apathy group. These results may help us to understand the neural alteration associated with apathy. This reviewer has several concerns as below.

Comment: The description of data preprocessing on page 8 is not clear. The phrase of ‘scrubbing parameter with a band-pass filter of 0.01-0.08 Hz’ is confusing. Considering the later sentence stated that the scrubbing parameter was used to identify the outlier scans, did the authors want to express in the first sentence that a band pass filtering was used and a scrubbing strategy was used to cutoff outlier scans using a ‘parameter’ generated by the toolbox? If so, what parameter and threshold was used to determine an outlier scan? For example, the frame-displacement (FD > 0.5 mm) was used in many previous studies.

Response: Thank you for your comment. We fixed the description as follows:

Page 8 Line 129

…, the scrubbing outlier scans were identified based on ART with intermediate settings (97 percentiles in normative sample), and band-pass filtering of 0.08–0.09 Hz.

supplementation: The number of the band-pass filter was wrong and we made the necessary modifications.

Comment: The description of “functional connectivity analysis” is not clear. It was said that ‘In the first-level analysis, a multiple regression analysis was performed for each individual subject using the general linear model for correlation connectivity estimation’. My understanding of the procedure is that ‘the multiple regression analysis’ was used in the preprocessing stage to regress out nuisance variables including head motions and others. Then the resultant errors were band-pass filtered. The time course of each seed mask was extracted and correlated with the time course of each voxel. The correlation coefficients were then transferred with Fisher’s Z-transformation.

Response: Thank you for this comment. Your summarization is true. We fixed the description as follows:

Page 9 Line 141

Data analyses were carried out using the CONN toolbox. In the first-level analysis, we obtained ROI-to-voxel functional connectivity maps for each individual subject. This analysis produced z-score maps of positive and negative correlation coefficients between 6 striatal seeds and the whole-brain voxels, combining homologous left and right ROIs to one seed. We also conducted an ROI-to-ROI analysis to examine functional connectivity between 12 ROIs within the striatum for each subject.

Comment: The FWHM of 8 mm seem too big as the seed ROIs are close to each other. In literature, 5 or 6mm is a typical FWHM and may be more proper for the current study.

Response: Thank you for mentioning this important point. The reason for using the large FWHM in the present analysis is that the interest of this analysis is in the functional connectivity of the frontal robe. It is known that functional connectivity in the frontal robe shows large individual differences (Mueller et al., 2013). Therefore, we decided that 8-mm is appropriate enhance sensitivity of the analysis in this study. To confirm this, we conducted the additional analysis using the data smoothed with 6 mm FWHM. As a result, the right dorsal anterior cingulate cortex/ pre supplementary motor area, which showed significant group difference with the 8 mm FWHM, did not meet the criteria for significance with 6 mm FWHM. Thus, a comparably wide FWMH may be useful when investigating functional connectivity of the frontal lobe with the basal ganglia as a seed, due to the large individual variability.

Comment: The left and right homologous ROIs were combined to one seed when calculating functional connectivity maps. But the human brain laterization is well known. It’s better to conduct a 2-by-2 ANOA for the second level analysis with GROUP and HEMISHPERE as factors.

Response: Thank you for this advice. Follow the reviewer’s suggestion, to check the effect of apathy on laterality of functional connectivity between basal ganglia and cerebral cortex, we conducted a two-way ANOVA with GROUP (apathy/non-apathy) and HEMISHPERE (the bilateral pairs of seeds), and the result did not show significant difference between right and left striatum seeds. We have added this result to our Results section as follows.

Page 12 Line 185

To investigate the effect of apathy on the laterality of functional connectivity between the basal ganglia and cerebral cortex, we conducted a two-way ANOVA with GROUP (apathy/non-apathy) and HEMISHPERE (the bilateral pairs of seeds), and the result did not show significant difference between the right and left striatum seeds.

Comment: When reporting results, the functional connectivity of several seeds seems to have no difference between the two groups. But it was not explicitly stated.

Response: Thank you for this comment. We fixed the description as follows:

Page 12 Line 184

There were no significant group differences in the analyses of other seeds.

Comment: The readability of Figure 2b is poor. I suggest a different font color or change the background color.

Response: Thank you for this advice. We changed the background color of Figure 2b accordingly.

Reviewer #.2:

Major points:

Q1. The areas associated with apathy in AD are not just the ACC and OFC, or the entire ACC or OFC. If it is desired to exemplify some of the related brain regions, it may be more appropriate to reconstruct the sentence (Introduction section- Line 26).

A1. The reviewer is correct and we have corrected the sentence appropriately as follows:

In Alzheimer's disease, apathy is associated with the atrophy of several brain regions, including the anterior cingulate cortex (ACC) and orbitofrontal cortex (OFC)[3].

supplementation: About the third reference, we replaced the original reference, since we were citing a different reference previously.

Q2..7th reference seems to be missing in the main text.

A2. As indicated, the 7th reference has been added in the main text.

Q3. Extended sentences with conjunctions make it difficult to follow, and in some there may be errors in tense suffixes. In this sense, I would suggest that the manuscript be reviewed by the authors.

A3. We revised the entire paper and tried to use the shortest sentences possible. Additionally, although we hired a proofreading service prior to the initial submission of our manuscript, we have had our manuscript rechecked for publication. We will attach a certification for the resubmission.

Q4. The authors may consider to edit the flow of Introduction section. The paragraph starting with "Apathy is also observed in older adults with..." from line 43 does not seem to be compatible with the previous paragraph. In the previous paragraph, the conceptualization of the apathy and the functional neuroanatomical correlates corresponding to this conceptualization are presented. However, the first sentence containing the word "also" causes the expectation for continuum of a topic.

A4. Thank you for this comment. In the previous version, we used the word “also” to indicate that healthy older adults experience apathy as well, but this was not clear. We fixed the description as follows:

Page 4 Line 44

Apathy is observed in older adults with normal cognitive function and its prevalence is increased with aging and reduces their quality of life.

Q5. I think, the first paragraph of the Materials and Methods/Participants section also need to be reconsidered. Especially these two sentences are hard to follow: "335 people in the database were screened on the levels of apathy and the exclusion criteria. Exclusion criteria were: current or past presence of neurological or psychiatric disorders such as cerebrovascular disease, Mini-mental State Examination (MMSE) score less than 26 or dementia, diffuse or multiple cerebral white matter lesions on T2-weighted image, obvious cerebral atrophy, head motion > 2.0 mm during the scan of rs-fMRI". Since this section contains basic information for readers to understand the study, I think it would be more useful to give it in simpler sentences.

A5. Thank you for this advice. We have modified these sentences as follows:

Page 6 Line 86

A total of 335 people in the database were screened on their apathy levels and the exclusion criteria. Exclusion criteria were: current or past presence of neurological or psychiatric disorders; Mini-mental State Examination (MMSE) score less than 26 or dementia; obvious structural brain disorders; and head motion > 2.0 mm during the scan of rs-fMRI.

Q6. It is not common to use cut-off points on apathy scales. In general higher scores indicate higher levels of apathy. Was the cut-off point for the apathy scale used in a single study? Are there any other examples in recent studies, supporting the validation of this cut-off score other than the relevant reference (#32)?

A6. Thank you for this comment. Reference #28 was wrong. Okada et al. performed statistical validation of the cut-off point for the apathy scale. However, there are several studies that use this criterion (e.g., Yan et al.2015, Sugawara et al. 2011). The study by Yan et al. detected atrophy of the motor cortex in elderly individuals with apathy, while the study by Sugawara et al. indicated that hearing impairment was significantly associated with both MMSE and AS scores using the same criteria.

References

Yan H, Onoda K and Yamaguchi S (2015) Gray matter volume changes in the apathetic elderly. Front. Hum. Neurosci. 9:318. doi: 10.3389/fnhum.2015.00318

Norio Sugawara et al. Hearing impairment and cognitive function among a community-dwelling population in Japan. Annals of General Psychiatry 2011, 10:27

Q7. Explaining the scoring of depression scale will make it possible to understand the results and the table. Does this depression scale involves any cognitive tasks? If not, I would strongly recommend not to cluster depression scale under the neuropsychological assessment.

A7. Thank you for making this important point. The depression scale does not include cognitive tasks. Therefore, the heading of this paragraph has been revised as follows:

Page 7 Line 97

Neuropsychological and Neuropsychiatric measures

Q8. Line 151. Giving an analysis result and comparative reference information in the Method section may not be very appropriate. I recommend to relocate that sentence to Discussion.

A8. Thank you for this comment. The analysis the reviewer pointed out was simply conducted to confirm the reproducibility of our study. We moved this analysis result to a different section of the Results as follows:

Page 11 Line 173

The functional connectivity map of 6 striatal ROIs across 36 participants was similar to a previous study (Di Martino et al.) that demonstrated connectivity between striatum subdivisions and other cerebral regions such as the frontal cortex, parietal cortex, and temporal cortex.

Q9. Under the figure: A)… Red and blue foci indicate areas showing significant positive and negative connectivity.

It is better to clarify the color and connectivity direction.

A9. Thank you for this recommendation. We modified this as follows:

Page 13 Line 204

Red and blue foci indicate areas showing significant positive and negative connectivity differences, respectively.

Q10. Creating a table with peak Talairach coordinates for ROIs can make it easier for the readers to understand the results. It will also be convenient for the study to be repeated by other researchers.

A10. Thank you for this advice. We created a new Table 2 accordingly.

Page 12 Line 196

Q11. The authors drew attention to the overlap in symptoms of apathy and depression. However, if apathy and depression comorbidly present; it may not be easy to draw a line between these two syndrome as we can do between an internal disease and a psychiatric condition. It is known that both depression and apathy are the neuropsychiatric conditions that have independent effects on the molecular structure and physiological functioning of the brain. Statistically controlling the depression score may not result with subtracting the effect of depression from the brain activation as intended here in this study. I find the study's functional imaging analysis approach correct (controlling depression by adding the scores to analysis as covariant). However, adding multi covariates in fMRI analysis may suppress the overall activation which can cause underestimating the findings. In my opinion, in order to be able to say that the results were clearly independent of depression, a group of apathetic participants without depression should have been included. I would recommend the authors to discuss their method and findings stronger, in this context. Also, if the researchers have used any other tools or measurements, those can be included to manuscript in terms of increasing internal consistency.

A11. Thank you for providing these insights. Regarding the part of your comment which states “… to be able to say that the results were clearly independent of depression, a group of apathetic participants without depression should have been included.” , we tried to pick such individuals up from our database. However, we could not find elderly patients with high apathy and no depression due to their high comorbidity. Therefore, in this revised manuscript, we have added this as a limitation of our study. Please confirm that this is an acceptable change.

Page 17 Line 282

Moreover, to ensure that the participants were clearly independent of depression, an apathy group without depression should have been included.

Minor points:

Q1. Line 41. Mistyping of striatum as stratum

A1. As indicated, the spelling has been corrected.

Q2. Line 60. I would suggest using "study of cognitively normal aged people" instead of "study of cognitive- normal aged people"

A2. As indicated, the term has been corrected.

Q3. Line 85. I would suggest using "cranial MRI" instead of "head MRI"

A3. The suggested term has been used.

Q4. Line 104. This long sentence "Furthermore, all participants conducted the following neuropsychological assessments: Cognitive functions, assessed by the MMSE [29], the Frontal Assessment Battery (FAB) [30], and the Kohs Block Design Test (KOHS) [31]; depressive symptoms were evaluated using the Japanese version of Zung’s self-rating depression scale (SDS) [32]." can be simply expressed like "Furthermore, all participants conducted the following neuropsychological assessments: MMSE [29], Frontal Assessment Battery (FAB) [30], and Kohs Block Design Test (KOHS) [31]. Depressive symptoms were evaluated using the Japanese version of Zung’s self-rating depression scale (SDS)"

A4. Thank you for your comment. We modified this sentence as follows:

Page 7 Line 104

Furthermore, all participants conducted the following neuropsychological assessments: MMSE [29], Frontal Assessment Battery (FAB) [30], and Kohs Block Design Test (KOHS) [31]. Depressive symptoms were evaluated using the Japanese version of Zung’s self-rating depression scale (SDS) [32].

Q5. Line 126. Could the authors have meant "adjust" by using the phrase "to agree with" in this sentence: Next, the functional images were normalized to the standard space defined by a template T1-weighted image and resliced with a voxel size of 3x3x3 mm3 to agree with the gray matter probability maps.

A5. Thank you for your advice. We made modifications as follows:

Page 8 Line 123

Next, the functional images were normalized to the standard space defined by a template T1-weighted image and resliced with a voxel size of 3x3x3 mm3 to adjust the gray matter probability maps.

Q6. Line 209. If the intended meaning is not the process of aging; authors may consider using "elderly" instead of using the expression "aged people".

A6. The suggested expression has been used as follows:

Page 13 Line 214

Our results indicate that functional disruption of frontal-striatal circuits is associated with apathetic state in the elderly, independent of depression.

We look forward to hearing from you regarding our submission. We would be glad to respond to any further questions and comments that you may have.

Sincerely,

Chizuko Hamada, M.D.

Department of Neurology, Shimane University

89-1, Enyacho, Izumo, Shimane, Japan, 693-8501

E-mail: okuzihc@med.shimane-u.ac.jp

---

## [Decision Letter · Decision Letter 1]

11 Nov 2021

PONE-D-21-12168R1Altered resting-state functional connectivity of the frontal-striatal circuit in elderly with apathyPLOS ONE

Dear Dr. Hamada,

Thank you for submitting your manuscript to PLOS ONE. After careful consideration, we feel that it has merit but does not fully meet PLOS ONE’s publication criteria as it currently stands. Therefore, we invite you to submit a revised version of the manuscript that addresses the points raised during the review process.

Please address the comment of reviewer #1.   

We look forward to receiving your revised manuscript.

Kind regards,

Satoshi Ikemoto

Academic Editor

PLOS ONE

Journal Requirements:

Reviewers' comments:

Reviewer's Responses to Questions

**Comments to the Author**

1. If the authors have adequately addressed your comments raised in a previous round of review and you feel that this manuscript is now acceptable for publication, you may indicate that here to bypass the “Comments to the Author” section, enter your conflict of interest statement in the “Confidential to Editor” section, and submit your "Accept" recommendation.

Reviewer #1: All comments have been addressed

Reviewer #2: All comments have been addressed

2. Is the manuscript technically sound, and do the data support the conclusions?

Reviewer #1: Yes

Reviewer #2: Partly

3. Has the statistical analysis been performed appropriately and rigorously? 

Reviewer #1: Yes

Reviewer #2: Yes

4. Have the authors made all data underlying the findings in their manuscript fully available?

Reviewer #1: Yes

Reviewer #2: No

5. Is the manuscript presented in an intelligible fashion and written in standard English?

Reviewer #1: Yes

Reviewer #2: Yes

6. Review Comments to the Author

Reviewer #1: The authors have addressed most of my concerns well. However, in line 139, the bandpass filter COULD NOT be 0.08-0.09Hz, the typic one should be 0.01-0.09Hz. If this work was done with 0.08-0.09Hz, I do not think these is any signal left for functional connectivity calculation. Please Check.

Reviewer #2: Recommendations were found to be considered by the authors. According to my opinion, with the contributions of the editor and the other referee, this valuable study has become better presented. Since the significance intervals of the scores from the depression scale has not been explained, I think it is still not clear how pure is the patient group with apathy in the diagnostic sense. However, the authors were also showed sensitive approach to this point and took the recommendation into account by reporting this as a limitation. In this sense, there is nothing else I would like to point out. I hope that the authors will continue their work on this topic and replicate the results in further studies.

7. PLOS authors have the option to publish the peer review history of their article (what does this mean?). If published, this will include your full peer review and any attached files.

Reviewer #1: No

Reviewer #2: No

---

## [Author Response · Author response to Decision Letter 1]

29 Nov 2021

Journal requirements:（2021/11/12）

Q. Please review your reference list to ensure that it is complete and correct. If you have cited papers that have been retracted, please include the rationale for doing so in the manuscript text, or remove these references and replace them with relevant current references. Any changes to the reference list should be mentioned in the rebuttal letter that accompanies your revised manuscript. If you need to cite a retracted article, indicate the article’s retracted status in the References list and also include a citation and full reference for the retraction notice.

A. We appreciate the helpful suggestion regarding Journal requirements. We fixed our reference list as follows:

Page 24 Line 399-400

2019; 101(1):165-177.e5. doi:10.1016/j.neuron.2018.11.016

Page 24 Line 409

doi:10.31887/DCNS.2016.18.1/shaber

Page 25 Line 414

doi:10.1523/JNEUROSCI.11-03-00667.1991

Page 25 Line 428

doi:10.1016/sS0006-8993(99)01531-0

Comments to the Author

6. Review Comments to the Author

Reviewer #1

Q. The authors have addressed most of my concerns well. However, in line 139, the bandpass filter COULD NOT be 0.08-0.09Hz, the typic one should be 0.01-0.09Hz. If this work was done with 0.08-0.09Hz, I do not think these is any signal left for functional connectivity calculation. Please Check.

A. Thank you for your comment. We have confirmed, and the bandpass filter was 0.008, not 0.08. We fixed the bandpass filter as follows:

Page 8 Line 130

band-pass filtering of 0.008–0.09 Hz.

We look forward to hearing from you regarding our submission. We would be glad to respond to any further questions and comments that you may have.

---

## [Editor Report · Decision Letter 2]

1 Dec 2021

Altered resting-state functional connectivity of the frontal-striatal circuit in elderly with apathy

PONE-D-21-12168R2

Dear Dr. Hamada,

We’re pleased to inform you that your manuscript has been judged scientifically suitable for publication and will be formally accepted for publication once it meets all outstanding technical requirements.

Kind regards,

Satoshi Ikemoto

Academic Editor

PLOS ONE
---

## [Editor Report · Acceptance letter]

3 Dec 2021

PONE-D-21-12168R2 

Altered resting-state functional connectivity of the frontal-striatal circuit in elderly with apathy 

Dear Dr. Hamada:

I'm pleased to inform you that your manuscript has been deemed suitable for publication in PLOS ONE. Congratulations! Your manuscript is now with our production department. 

Kind regards, 

on behalf of

Dr. Satoshi Ikemoto 

Academic Editor

PLOS ONE